# Discovery of Hepatotoxic Equivalent Markers and Mechanism of *Polygonum multiflorum* Thunb. by Metabolomics Coupled with Molecular Docking

**DOI:** 10.3390/molecules28010025

**Published:** 2022-12-21

**Authors:** Yinhuan Zhang, Lirong Liu, Menghan Feng, Hao Wu, Yihang Dai, Zhixin Jia, Cong Fang, Mingyan Liu, Xiaoning Yan, Meixia Zhu, Beibei Huang, Biqiong Qu, Hongbin Xiao

**Affiliations:** 1School of Chinese Materia Medica, Beijing University of Chinese Medicine, Beijing 100029, China; 2Research Center of Chinese Medicine Analysis and Transformation, Beijing University of Chinese Medicine, Beijing 100029, China; 3Beijing Research Institute of Chinese Medicine, Beijing University of Chinese Medicine, Beijing100029, China

**Keywords:** *Polygonum multiflorum* Thunb., hepatotoxic equivalent markers, liver injury, metabolomics, hepatotoxic mechanism

## Abstract

*Polygonum multiflorum Thunb*. (PMT), a commonly used Chinese herbal medicine for treating diseases such as poisoning and white hair, has attracted constant attention due to the frequent occurrence of liver injury incidents. To date, its hepatotoxic equivalent markers (HEMs) and potential hepatotoxic mechanisms are still unclear. In order to clarify the HEMs of PMT and further explore the potential mechanisms of hepatotoxicity, firstly, the chemical constituents in PMT extract were globally characterized, and the fingerprints of PMT extracts were established along with the detection of their hepatotoxicity in vivo. Then, the correlations between hepatotoxic features and component contents were modeled by chemometrics to screen HEMs of PMT, which were then further evaluated. Finally, the hepatotoxic mechanisms of PMT were investigated using liver metabolomics and molecular docking. The results show that the chemical combination of 2,3,5,4-tetrahydroxystilbene-2-O-β-D-glucoside (TSG) and emodin-8-O-glucoside (EG) was discovered as the HEMs of PMT through pre-screening and verifying process. Liver metabolomics revealed that PMT caused liver injury by interfering with purine metabolism, which might be related to mitochondrial function disorder and oxidative injury via the up-regulations of xanthosine and xanthine, and the down-regulation of 5′ nucleotidase (NT5E) and adenylate kinase 2 (AK2). This study not only found that the HEMs of PMT were TSG and EG, but also clarified that PMT might affect purine metabolism to induce liver injury, which contributed to our understanding of the underlying mechanisms of PMT hepatotoxicity.

## 1. Introduction

*Polygonum multiflorum* Thunb. (PMT), the root of *Polygonum multiflorum*, a member of Polygonaceae, is an important traditional Chinese medicine, and it has many potential uses such as detoxification, carbuncle elimination, bowel relaxation, sore scabies, ringworm, pruritus, malaria prevention, antiaging, eliminating dampness, reducing lipid and hair blackening [1], which have attracted serious interest due to the frequent occurrences of liver injury incidents. Clinical cases have reported that PMT could cause liver toxicity to different degrees, most of which are related to long-term use and overdoses of drugs [2,3,4], as well as idiosyncratic hepatotoxicity [5,6,7,8]. Studies in vivo and in vitro have also demonstrated that PMT could induce cell injury, cholestasis and idiosyncratic liver injury [9,10,11,12,13]. There are many phytochemicals in PMT, including stilbenes, anthraquinones, glycosides, phospholipids, flavonoids, tannins, fatty acids, alcohols, aldehydes, and ketones [14,15]. Current thinking has indicated that anthraquinones, stilbenes, tannins and their derivatives may be responsible for the hepatotoxicity of PMT [8,16,17,18,19,20,21]. However, it is well-known that herbal medicines consist of plenty of complex phytochemicals. The toxicity of PMT may be due to the chemical combination rather than a single constituent, hence the hepatotoxic components of PMT remain controversial and the authentic hepatotoxic equivalent markers (HEMs) representing holistic hepatotoxicity of PMT urgently need to be clarified.

Metabolomics, regarding the body as a complete system with the characteristics of integrity and systematization [22], has been widely used to study the pharmacological and toxicological mechanisms of traditional Chinese medicine [23,24]. So far, serum, liver, plasma, bile acid and urine samples have been analyzed using untargeted metabolomics methods integrated with pattern recognition to investigate the potential hepatotoxicity of PMT, and 16 endogenous metabolites in serum [10], 10 metabolites in liver tissues [25], 16 highly specific biomarkers in plasma [26], 40 metabolites in liver slices [27], and 16 [28] and 12 [29] biomarkers in urine samples have been identified, respectively, which are involved in amino acid, fatty acid, bile acids, lipids, citrate cycle, vitamin B6, bilirubin, tryptophan, bile acid synthesis, purine metabolism, fatty acid oxidation and the energy metabolism. Additionally, the glycodeoxycholic acid in bile and hyodeoxycholic acid [30] in serum, as well as the Cer (d18:1/24:1), dihydroceramide (d18:1/18:0)-1-phosphate (dhCer (d18:1/18:0)-1P) and Cer (d18:1/26:1) [31] in plasma were assigned as potential hepatotoxic biomarkers for PMT by targeted metabolomics. Furthermore, 21 potential biomarkers related to the idiosyncratic hepatotoxicity of PMT were determined, which were mainly associated with the tricarboxylic acid cycle and sphingolipid metabolism pathways [32]. All the above results have provided preliminary information on mechanisms of liver injury induced by PMT. However, the relationships between metabolites, targets and metabolic pathways have not been elucidated, which has led to incomplete explanations of the hepatotoxic mechanism of PMT.

Metabolites participate in a wide variety of biochemical processes by interacting with various proteins. It has been found that metabolite regulators can interfere with enzyme activity by affecting metabolite–protein interactions, thereby affecting biochemical processes [33,34]. Enzyme activity is modulated by different concentrations of various metabolites, thereby allowing different regulations of enzyme activity [35]. Integrating information about metabolite–protein interactions can provide a better understanding of the regulatory networks and connections in biological pathways. In recent years, a number of highly sensitive methods have been developed for drug–protein interactions, which can also be applied to the analysis of metabolite–protein interactions. With advances in computer science, in silico tools are becoming increasingly popular, such as docking [36]. These methods utilize computational schemes to determine the docking configurations of a metabolite’s binding sites on proteins, providing information about the structure of a metabolite–protein complex at a given binding site, including total binding free energy, internal protein cavities, binding sites, the extent of motions and conformational changes within the protein interaction sites, solvent accessibility of binding residues, root mean square deviation, configurational entropy values and hydrogen bond networks [37,38,39,40,41,42]. Among them, docking is a powerful tool to allow accurate structural modeling and the prediction of activities between small molecules and the binding sites of target proteins through proper search algorithms and scoring functions based on the structure of the binding site [43]. This method is rapid and has a high success rate. Most available docking programs can achieve a prediction success rate of 70–80% with an accuracy of 1.5 to 2 Å [44].

In summary, on the one hand, the research strategy of combining fingerprints with chemometrics [45] was adopted to identify HEMs responsible for the overall hepatotoxicity of PMT. On the other hand, focusing on targets regulated by PMT and interacting with metabolites, we combined metabolomics with molecular docking to explore the mechanism of PMT-induced liver injury. In light of the above results, this study aimed at clarifying the hepatotoxic material basis and mechanism of PMT, which might provide a theoretical basis for its applications and development.

## 2. Results

### 2.1. Fingerprint Analysis of PMT Extracts

After preparation, the average yield of 10 batches of PMT extract was 31.82 ± 4.39%. Firstly, in order to validate the developed method, the analytical methodology was verified. The results show the relative standard deviations of precision, stability and repeatability were no more than 9%, which indicates that the analysis was reliable and repeatable, and this analytical method was applicable to the similarity and quantitative analysis of different batches of PMT samples. Then, the fingerprints of PMT extracts were established by UPLC (Figure 1A). Twelve common peaks were defined in the fingerprint, which accounted for above 80% of the overall peak areas. The similarity results of 10 batches of PMT samples are presented in Table 1, and the similarity values between the fingerprints of 10 batches of PMT samples and reference fingerprint fell within the range of 0.982–0.998. The results show that the fingerprints of 10 batches of samples were highly similar to the reference fingerprint, and the quality of PMT was stable.

### 2.2. Hepatotoxicity of PMT Extracts

The levels of serum ALT and AST activities were used to reflect liver function, and obvious elevations in these indexes reflect damage of the hepatic function [46,47]. To assess the hepatotoxicity of PMT extract, serum ALT and AST in mice treated with 10 batches of PMT extracts were detected. The results exhibit that the levels of serum ALT and AST in the model group underwent no significant changes when compared to the normal control group (ALT, *p* > 0.05; AST, *p* > 0.05), demonstrating that idiosyncratic mice were successfully established. It was noted that serum AST and ALT levels were significantly increased in the PMT extract group (*p* < 0.05), compared with the model group and normal group (Figure 1B). In addition, liver histopathological observations show that the morphology of hepatocytes around the central vein of liver lobules in the normal control group was clear and normal, without swelling or other types of cytopathic inflammatory cell infiltration. There was slight inflammatory cell infiltration in the model group, and mice in different batches treated with PMT showed disordered hepatocytes that were characterized by different degrees of hepatic cell necrosis and inflammatory cell infiltration (Figure 1C). These findings indicate that mice administered with PMT extracts at 5.46 g/kg showed potential hepatotoxicity.

### 2.3. Discovery of Hepatotoxic Markers

In the present study, we tried to establish a fingerprint–hepatotoxicity relationship model using the chemometric method to screen the possible hepatotoxic markers from PMT extracts [48,49]. Here, taking serum ALT and AST values as y values and the peak areas of 12 components in 10 batches of PMT as the χx variable, the OPLS model was established using the SIMCA software. The intercept of the R^2^ on the y-axis was 0.0227, less than 0.4. The intercept of Q^2^ on the y axis is −0.107, less than 0.05, indicating that the model had not been over-fitted (Figure 2A), which indicates that OPLS was appropriate for use in establishing a fingerprint–hepatotoxicity correlation model. The VIP value represents the contribution of each variable to the projection. Generally, when the VIP value was larger, the contribution was greater, where VIP values with a threshold > 1.0 indicate the greatest influence on the model [50]. As Figure 2B shows, the VIP values of peaks 4 and 8 were higher than 1.0 (3.206 and 1.199, respectively). By comparing the retention time with reference standards, peaks 4 and 8 were unambiguously confirmed to be TSG and EG, respectively (Figure 2C), indicating that TSG and EG might induce liver injury.

Furthermore, the relationships between the contents of TSG and EG and the activities of ALT and AST in 10 batches of PMT are shown in Figure 2D. The high contents of TSG and EG in samples S4, S8, and S10 are consistent with the high levels of ALT and AST, while in batches S1, S6, and S7, the low levels of TSG and EG were associated with their low activity in ALT and AST, which reveals that higher contents of TSG and EG were linked to higher hepatotoxicity.

Finally, the correlation analysis was carried out and the results are shown in Table 2; the spearman correlation coefficients of the contents of TSG, EG, TSG + EG and the activity of AST were 0.7552, 0.6868 and 0.7482, respectively, after the correlation analysis with GraphPad Prism 8.3.0, which shows that there was a strong and significant correlation (*p* = 0.0115, *p* = 0.0283, *p* = 0.0128) between the contents of TSG, EG and TSG + EG and the activity of AST. However, the contents of TSG, EG and TSG + EG and serum ALT activity had no significant associations (*p* > 0.05).

By summing up the results from the OPLS and correlation analyses, TSG and EG were finally discovered to be the main hepatotoxic compounds in the PMT extract. Accordingly, these two compounds were tentatively assigned as candidate HEMs accounting for the whole hepatotoxicity of the original PMT extracts.

### 2.4. Assessment of Hepatotoxic Equivalence between Candidate HEMs and Original PMT Extracts

In the hepatotoxic equivalence assessment, both original PMT extracts and the candidate TSG and EG mixture in S2 batch caused significant elevations in serum ALT and AST (*p* < 0.001–0.01) (Figure 3A); histopathologic analysis revealed inflammatory cell infiltration in the livers of mice given original PMT extracts and candidate markers (Figure 3B). The results show that the hepatotoxicity of the mixture of TSG and EG was almost equal to the hepatotoxicity of the PMT extract. The 90% CI values for the hepatotoxicity of candidate markers in the ALT and AST tests were 82.66–117.34% and 92.45–107.55%, respectively, which are within the range of 69.84~143.19%. The results show that TSG and EG might be the HEMs of PMT.

### 2.5. Liver Metabolomics Analysis

#### 2.5.1. Multivariate Statistical Analysis

To reveal the mechanism of liver injury caused by PMT, metabolomics was used to detect the effect of PMT on the metabolism of endogenous substances in the livers. PCA analysis showed that under the positive and negative modes, the sample enrichment trends of the normal control group, the model group, and the PMT group were distinguished (Figure 4A,B), which indicate that PMT induced remarkable changes in liver endogenous metabolites.

OPLS-DA analysis was used to better understand the different metabolic patterns and to identify potential metabolites that were significantly changed between the model and PMT groups (Figure 4C,F). The R^2^Y (cum) and Q^2^ (cum) of OPLS-DA in the positive model were 0.996 and 0.902, respectively, and 0.999 and 0.985 in the negative model, respectively, using the data from the model and PMT. These data indicate that models could be used for accurate predictions. Figure 4D,G show the S-plot diagram in positive and negative modes. To verify the validity of the multivariate analysis model, permutation tests with 200 iterations were further carried out. The validation plots indicate that the original models were valid in the positive (Figure 4E) and negative ion modes (Figure 4H).

#### 2.5.2. Identification of Potential Metabolite of PMT

Metabolites that meet a threshold of *p* < 0.05, VIP > 1.0 and |*p*(corr)| ≥ 0.58 were considered potential metabolites according to OPLS-DA analysis. Under these conditions, 17 endogenous metabolites were identified with 7 up-regulated and 10 down-regulated after exposure to PMT (Table 3). In addition, in order to intuitively reflect the relative changes in different metabolites in each group, MetaboAnalyst 5.0 was used to construct a heat diagram between 17 potential biomarkers of liver samples in different groups (Figure 4I).

As shown in Table 3 and Figure 4I, compared with the model group, PMT markedly increased the relative expressions of seven metabolites, including D-Malic acid, 3-(Imidazol-4-yl)-2-oxopropyl phosphate, xanthosine, xanthine, p-Cresol sulfate, 3-Carbamoyl-2-phenylpropionaldehyde and LysoPE(0:0/24:6(6Z,9Z,12Z,15Z,18Z,21Z)). Moreover, the expressions of 10 differential metabolites were significantly down-regulated by PMT, including hypoxanthine, propionylcarnitine, benzocaine, LysoPC(0:0/16:0), 3-[(3-Methylbutyl)nitrosoamino]-2-butanone, fexofenadine, cytosine, 5-Methylthiopentanaldoxime, LysoPE(0:0/20:4(5Z,8Z,11Z,14Z)) and leukotriene D4. The results show that the liver injury induced by PMT was related to the disturbance of these metabolites.

#### 2.5.3. Pathway Analysis

MetaboAnalyst 5.0 was applied to reveal the metabolic pathways of potential metabolites [51,52,53] related to PMT. Three pathways were enriched: purine metabolism, drug metabolism–cytochrome P450, and arachidonic acid metabolism (Figure 5A and Table 4). Among them, according to *p* < 0.05, the liver damage of immune-specific mice induced by PMT was mainly related to purine metabolism.

### 2.6. “Metabolite−Target−Pathway” Interactive Network Analysis

Based on the keywords of “liver injury” and “hepatotoxicity”, 6194 targets related to liver injury were screened. According to *p* < 0.05, purine metabolism was considered to be the most closely related to liver injury in mice. Therefore, 134 targets related to the purine metabolism pathway were obtained from the KEGG database. To systematically clarify the complex relationships between potential biomarkers and their targets, the STITCH, SEA, PharmMapper, Swiss Target Prediction and Super-PRED databases were used for collecting specific metabolite-related targets. In total, 565 targets corresponding to three potential metabolites related to purine metabolism were collected. Using the Venn diagram, 15 overlapping targets of liver injury, metabolites and metabolic pathways were obtained (Figure 5B), including purine nucleoside phosphorylase (PNP), cytosolic purine 5′-nucleotidase (NT5C2), cytosolic 5′-nucleotidase 1A (NT5C1A), 5′ nucleotidase (NT5E), adenosine deaminase (ADA), adenylate kinase 2 (AK2), inosine 5′-monophosphate dehydrogenase 2 (IMPDH2) soluble calcium activated nucleotidase 1 (CANT1), xanthine dehydrogenase/oxidase (XDH), hypoxanthine guanine phosphoribosyltransferase (HPRT1), guanine deaminase (GDA), cGMP inhibited 3′,5′-cyclic phosphodiesterase A (PDE3A), cGMP specific 3′,5′-cyclic phosphodiesterase (PDE5A), adenosine kinase (ADK), and exonucleotide pyrophosphatase/phosphodiesterase family member 1 (ENPP1). Figure 5C showed the interactions between the 15 overlapping targets. Cytoscape was used to visualize three metabolites, 15 overlapping targets and the metabolic pathways related to purine metabolism in PMT (Figure 5D).

### 2.7. Molecular Docking

In order to further evaluate the interactions between overlapping targets and related metabolites, three differential metabolites (hypoxantine, xanthosine and xanthine) were docked to 14 key targets (ADA: 3IAR [54,55]; ENPP1: 4B56 [56,57,58,59]; ADK: 4O1L [60,61]; PDE5A: 5JO3; PDE3A: 7L27; XDH: 2CKJ [62]; NT5E: 4H1S; PNP: 1M73; NT5C2: 2J2C [63]; AK2: 2C9Y [64]; IMPDH2: 6I0O; CANT1: 2H2N; HPRT1: 1Z7G; GDA: 2UZ9 [65]; NT5C1A: no protein structure temporarily). The results were listed in Table 5. Xanthosine showed good interactions with NT5E and AK2 based on the total score ≥ 5. The higher the similarity value, the more similar the molecular conformation was. Generally, a similarity value greater than 0.5 indicated that the molecular conformation was more similar. Meanwhile, the docking process was accompanied by the generation of collision force, which means there was a collision fraction crash. The higher the fraction of crash, the better the energy matching between the ligand molecule and the cavity formed by the protein’s active site. As shown in Table 5, the similarity values of NT5E and AK2 were 0.51 and 0.50, and the crash values of NT5E and AK2 were −0.6 and −0.98, respectively. In addition, hydrogen bonds are the most important specific interactions in biological recognition process. As shown in Figure 6, there were five hydrogen bonding forces between xanthosine and the residues ASN A: 77, PHE A: 294, GLU A: 169, ASP A: 168 and VAL A: 166 of the target NT5E. Xanthosine mainly had four hydrogen bonding forces with LEU A: 35, CYS A: 40, SER A: 228 and ASN A: 38 of target AK2. Then, auto-dock was further used to verify the combination of xanthosine with NT5E and AK2. The docking results (Table 6) showed that the binding energy of xanthosine with NT5E was −6.95 kcal/mol, and that of xanthosine with AK2 was −6.58 kcal/mol. Lower binding energy represented a more stable ligand–receptor interaction. The binding energy was lower than −5 kcal/mol, indicating that the docking results were stable, which further verified the results derived by the SYBYL-X software. These results showed that PMT might affect the interactions between xanthosine and NT5E and AK2 by up-regulating the expression of xanthosine, resulting in a high-order structural effect, thus inducing liver damage.

### 2.8. PMT Induced Oxidation Damage

In order to clarify the effect of PMT on the liver oxygen radical, the content of malondialdehyde (MDA), the activities of superoxide dismutase (SOD) and glutathione peroxidase (GSH-Px) were detected. The results were shown in Figure 7. There were no significant changes in MDA content, GSH-Px and SOD activities in the model group compared with the normal control group, indicating that LPS molding did not cause oxidative damage to the livers. Compared with the model group, PMT not only significantly inhibited the activities of GSH-PX and SOD, but also caused a significant increase in MDA content in the liver. These results suggested that PMT inhibited the antioxidant capacity, aggravated the rise of oxygen free radicals, and caused oxidative damage.

### 2.9. Western Blot Validation

The results were shown in Figure 8. Compared with the normal control group, the protein expressions of NT5E and AK2 in the model group showed no significant changes, indicating that LPS modeling had no effect on the protein expressions of NT5E and AK2. However, compared with the model group, when PMT was administered, the protein expressions of NT5E and AK2 in the liver were significantly down-regulated, indicating that PMT could remarkably inhibit the protein expressions of NT5E and AK2.

## 3. Discussion

Chemical profiling of herb medicine is a condition of the highest importance for excavating bioactive compounds [66,67]. Many studies have been attempted for discovering the hepatotoxic elements of PMT [6,68,69], while they have ignored the synergistic effects engendered by single components. Similarity analysis of chromatographic fingerprint is well-recognized as a useful measure to evaluate the batch-to-batch chemical consistency of Chinese herbs [70,71,72,73]. The chemical similarities among 10 batches of PMT extracts were measured via the fingerprint analysis. It was noted that all the similarity values were higher than 0.982, suggesting that the chemical fluctuation between samples was very little. Conversely, the hepatotoxic effects of PMT extracts highly varied. Therefore, based on the fingerprint–toxicity relationship model, which has been successfully applied in discovering bioactive or hepatotoxic components [45,74,75], OPLS was applied to screen hepatotoxic markers from PMT extracts. Two principal compounds, TSG and EG, were discovered as the hepatotoxic markers of PMT. Consequently, the combination of these two compounds was assigned as the candidate HEMs accounting for the whole hepatotoxicity of the original PMT extracts. As shown in Figure 2D, the sum of TSG and EG correlated well with hepatotoxicity. In samples S01, 06 and 07, low amounts of TSG and EG were associated with low serum ALT/AST levels. While in samples S04, 08 and 10, a similar trend was observed, relatively higher contents of TSG and EG corresponded to more potent toxicity. The content–toxicity correlation implies that the combination of TSG and EG might be a candidate HEM for the hepatotoxicity evaluation of PMT extract. Additionally, it is generally believed that the effective components of drugs are usually the prototype components or their metabolites, which exist in the blood or target organs. In a study, 23 components were detected in rat plasma samples after intragastric administration of the ethanol extract of raw PMT, including 16 prototype components such as emodin and EG [76]. TSG and EG were also detected in the livers of rats fed with raw PMT [77], which further indicated that TSG and EG could be used as the basis of liver toxicity of PMT. The hepatotoxicity equivalence between candidate HEMs and PMT extracts was further verified. It was found that the mixtures of TSG and EG showed nearly equivalent hepatotoxicity with corresponding PMT extracts, since all the 90% CI values fell within the range of 69.84−143.19% [78], demonstrating that candidate HEMs equaled the original PMT extracts in terms of hepatotoxicity. Hence, the chemical combination of TSG and EG was confirmed as HEMs for PMT.

Furthermore, we explored the hepatotoxic mechanism of PMT. Liver metabolomics analysis found that the mechanism of liver injury caused by PMT may be related to purine metabolism [27], which was consistent with our metabolomics results, where purine metabolism was the most significant pathway for PMT inducing liver injury. The productions of superoxide and H_2_O_2_ were accompanied by the decomposition and metabolism of purine molecules to produce uric acid, which mainly emerged in the process of the catalytic conversion of hypoxanthine to xanthine and the metabolism of xanthine to urate [79]. Our results showed that PMT influenced the purine metabolic pathway, and induced the up-regulation of xanthosine and xanthine, which likely contributed to the increase in oxygen radicals. Free radicals can react with unsaturated fatty acids to form the lipid peroxidation product MDA. SOD and GSH-Px are important antioxidant enzymes in the body, which can eliminate oxygen free radicals, reduce the formation of MDA, and protect biofilm and biological macromolecule structure. It was found that PMT could cause an increase in MDA content and a decrease in GSH activity in liver cells, and reduce the ability of scavenging oxygen free radicals [80]. Although the possibly unfavorable factor of hypoxanthine down-regulation existed, our results also proved that PMT inhibited the enzyme activities of SOD and GSH-Px in livers, and increased the content of MDA, indicating that PMT gave rise to the accumulation of free radicals and led to the oxidative stress damage of liver tissues. In addition, the excessive production of oxygen radicals, on the one hand, could directly induce inflammatory reactions, leading to the synthesis and release of inflammatory factors. On the other hand, inflammatory factors could promote the production of free radicals, thereby aggravating inflammatory reactions and forming a vicious circle. It was reported that PMT could cause immune inflammatory damage and significantly elevate the levels of plasma TNF-α, IL-1 β, IL-6 and IFN-γ in immune idiosyncratic rats [81].

It has been universally acknowledged that the interactions between proteins and small molecule metabolites play a vital role in regulating protein functions and controlling various cellular processes. Thus, in order to further comprehend the molecular mechanisms of liver injury, the targets of metabolites, liver injury and purine metabolic pathways were analyzed by synthesis. Our metabolomics analysis showed that purine metabolism was the most significant pathway for PMT inducing liver injury, and a total of 134 targets were involved in purine metabolism. Venn diagram analysis showed 15 overlapping targets related to liver injury, metabolites, and purine metabolism pathway, including PNP, NT5C2, NT5C1A, NT5E, ADA, AK2, IMPDH2, CANT1, XDH, HPRT1, GDA, PDE3A, PDE5A, ADK and ENPP1. This indicated that hypoxanthine, xanthosine and xanthine might cause liver injury by affecting these targets. Alternatively, molecule docking showed that xanthosine could dock stably with NT5E and AK2, indicating that xanthosine interacted most closely with NT5E and AK2, which might cause liver injury by affecting NT5E and AK2.

NT5E is a glycosyl-phosphatidylinositol-linked plasma membrane glycoprotein that is expressed on multiple cell types and in different tissues [82]. NT5E could hydrolyze extracellular adenosine monophosphate into adenosine and inorganic phosphate [83], which acted as an important regulator of tissue homeostasis and pathophysiologic responses related to immunity, inflammation, ischemia, tissue fibrosis, and cancer [84,85,86]. Specifically, the deficiency of NT5E in the liver of male mice caused obvious liver cell damage, promoting the expression of cytokines IL-1β and Tnf-α and more neutrophil infiltration [87]. The low expression of NT5E promoted the formation of the Mallory–Denk body [88]. CD73 mRNA was significantly reduced in patients with fibrosis caused by hepatitis C infection or nonalcoholic fatty liver disease [89]. These studies indicated that NT5E was closely associated with liver injury. Our experiment found that PMT inhibited the protein expression of NT5E, which might be the cause of liver injury. The adenosine produced by NT5E acted on adenosine receptors (AR) to activate downstream G protein-coupled signaling [90,91], such as MAP kinase (MAPK) [92]. AMPK regulated the reactive oxygen species (ROS) production and antioxidation of mitochondria. Cells lacking AMPK activity showed decreased expressions of antioxidant genes, including catalase, SOD2, and increased mitochondrial ROS levels [93]. It was reported that a deficiency of NT5E led to AMP-activated protein kinase hypoactivation and perturbs metabolic homeostasis, causing liver damage in mice [87]. Therefore, the NT5E protein was inhibited by PMT, which might cause the low expression of adenosine. Adenosine might cause the insufficient activation of AMPK, which in turn leads to the down-regulation of antioxidant SOD and GSH Px and an increase in MDA, resulting in oxidative stress and ultimately liver injury (Figure 9). Adenylate kinase is the critical enzyme in the metabolic monitoring of cellular adenine nucleotide homeostasis, which catalyzes the reaction 2ADP↔AMP + ATP; this is a recognized facilitator of AMP metabolic signaling, optimizing intracellular energetic communication and local ATP supply [94]. The adenylate kinase network provides an efficient mechanism for high-energy phosphoryl transport from mitochondria to ATP utilization sites [95]. Adenylate kinase 2 isoform AK2 is located in the intermembrane and intra-cristae space and facilitates high-energy phosphoryl exchange between mitochondria and cytosol [94]. AK2 deficiency disrupts cell energetics, [96] compromises the function of the mitochondrial respiratory chain [97] and causes the hyperpolarization of mitochondria and ROS production [98]. Our results showed that PMT inhibited the protein expression of AK2, which may further damage mitochondria, causing liver injury (Figure 9). Researchers also confirmed that PMT could reduce the functions of respiratory chain and mitochondrial membrane potential energy, causing disorders in the energy metabolism [80]. Moreover, with AMP as the substrate, AK2 and NT5E also participated in the generation of hypoxanthine. The low protein expressions of NT5E and AK2 caused by PMT might also reasonably explain the down-regulation of hypoxanthine in metabonomic data, which further verified the reliability of our metabolomics data. In general, PMT caused the pathological emergence of liver injury through purine metabolism, which might be related to NT5E and AK2, causing mitochondrial metabolism disorder and oxidative stress.

## 4. Materials and Methods

### 4.1. Chemicals and Materials

A total of 10 batches of PMT samples (S01–10) were collected from different regions in China. The voucher specimens which were identified by Professor Xueyong Wang of Beijing University of Chinese Medicine, and were deposited in the Research Center of Chinese Medicine Analysis and Transformation, Beijing University of Chinese Medicine, Beijing, China. The HPLC-grade solvents, acetonitrile, methanol and formic acid were purchased from Thermo Fisher Scientific (Waltham, MA, USA), and deionized water (18 MΩ cm) was prepared by passing distilled water through a Milli-Q system (Millipore, Milford, MA, USA). Other reagents and chemicals were of analytical grade. The reference standards of 2,3,5,4-tetrahydroxystilbene-2-O-β-D-glucoside (TSG, purity ≥ 98%) and emodin 8-O-glucoside (EG, purity ≥ 98%) were purchased from Chengdu Refines Biotechnology Co., Ltd. (Chengdu, China) and Beijing Tetrahedron Biotechnology Co., Ltd. (Beijing, China), respectively. Lipopolysaccharide (LPS) was purchased from Shanghai Yuanye Biotechnology Co., Ltd. (Shanghai, China). Rabbit Anti-NT5E Polyclonal Antibody was purchased from Bioss Biotechnology Co., Ltd. (Beijing, China). AK2 Polyclonal Antibody was purchased from Proteintech Group, Inc. (Chicago, IL, USA). Anti-β-actin was purchased from Abcam Technology (London, UK). Anti-rabbit IgG and horseradish peroxidase (HRP)-linked antibodies were purchased from Cell Signaling Technology (Boston, MA, USA). Prestained dual-color protein molecular weight markers for western blotting were obtained from Beyotime (Shanghai, China). Enhanced chemiluminescence (ECL) reagent was purchased from Merck Millipore (Darmstadt, Germany). The SDS gel preparation kit was purchased from Solarbio Biotechnology Co., Ltd. (Beijing, China).

### 4.2. Preparation of PMT Extracts

Different batches of raw PMT decoction pieces were crushed; 50 g of the powder was infiltrated 10 times with 60% ethanol, and then heated and refluxed for extraction twice, each for 1 h. The filtrate was suction-filtered while hot, and the filtrates were combined, which was then concentrated under reduced pressure and dried in vacuo subsequently. Then, 100 mg of PMT extract was dissolved in 25 mL 50% methanol and filtered through a 0.22 µm filter. The filtrate was injected for analysis.

### 4.3. Chromatographic and Mass Spectrometric Conditions

Chromatographic analysis was performed on an Agilent series 1290 UPLC system equipped with a quaternary pump, a degasser, a diode array detector and a thermo-stated column compartment (Agilent Technologies, Palo Alto, CA, USA). Chromatographic separation was carried out at 25 °C on an ACQUITY UPLC HSS T3 column (2.1 mm × 100 mm, 1.8 μm). The mobile phase was a mixture of 0.1% formic acid in water (A) and 0.1% formic acid in acetonitrile (B) with a gradient elution as follows: 0–2 min, 12–15% (B); 2–5 min, 15–18% (B); 5–18 min, 18–40% (B); 18–23 min, 40–65% (B); 23–29 min, 65–95% (B); 29–30 min, 95–95% (B). The flow rate was 0.4 mL/min, and column temperature was set at 35 °C. The detection wavelength was 254 nm and the injection volume was 5 μL.

### 4.4. Animal Experiments

Male ICR mice (18–20 g) were purchased from Beijing Weitonglihua Experimental Animal Technology Co., Ltd. (Beijing, China). The mice were fed a standard laboratory diet and given free access to tap water, kept in a controlled room temperature (20 ± 1 °C), humidity (60 ± 5%), and a 12/12 h light/dark cycle for at least one week before treatment. All animal experiments were approved by the Committee on Animal Care and Use of Institute of Beijing University of Chinese Medicine. The animal experiment license number was BUCM-4-2021082005-3050.

Mice were randomly divided into 12 groups, including the normal control group, model group and 10 batches of PMT administration group. With adaptive feeding for a week, they were fasted from food but not water for 12 h before the experiment. Mice in the model group and the administration group were intraperitoneally injected with 2 mg/kg LPS, and mice in the normal control were intraperitoneally injected with the same volume of normal saline. Two hours after modeling, 0.5% sodium carboxymethyl cellulose Na (CMC-Na) was used as a model control (*n* = 10) and mice in the administration group were orally administered PMT extracts (5.46 g/kg, suspended in 0.5% CMC-Na, *n* = 10) for 6 h. Blood was collected from the eyeballs to evaluate the biochemical indicators of liver function. Next, the liver tissues were collected for HE staining and metabolomics analysis.

### 4.5. Hepatotoxicity Analysis of PMT Extracts

Serum alanine aminotransferase (ALT) and aspartate aminotransferase (AST) activities were measured by biochemical analyzer (TBA-40FR, China). Liver tissues were fixed in 4% paraformaldehyde, paraffin-processed, and sectioned at 4 μm. For histological evaluation, the tissue sections were stained with hematoxylin and eosin (HE) and observed under the microscope.

### 4.6. Establishing of UPLC Fingerprint

All PMT samples were chemically profiled under the abovementioned chromatographic conditions. The fingerprints of 10 batches of samples (S01–S10) were matched automatically using the Similarity Evaluation System for Chromatographic Fingerprint of Traditional Chinese Medicine (2012 Version, Committee of Chinese Pharmacopeia). The simulative mean chromatogram as a representative standard for those fingerprints was calculated and generated automatically by the median method. Through the careful comparison of UV spectrum and relative retention time, the peaks detected in all fingerprints were defined as “common characteristic peaks”, and the similarity was evaluated.

To identify the fingerprint–toxicity relationships and find the contributions of each component in PMT to liver injury, the orthogonal partial least squares (OPLS) model was applied using SIMCA software (version 14.0.1, Umetrics AB, Umea, Sweden).

### 4.7. Quantitative Analysis of the Major Biomarkers

The contents of TSG and EG were quantified by external standard method using the above chromatographic conditions and the developed UPLC method, which was fully validated before quantification.

### 4.8. Hepatotoxicity Evaluation between Candidate HEMs and PMT Extracts

Hepatotoxic equivalence was evaluated by calculating the 90% confidence interval (CI) of the ratio between the toxicities of candidate HEMs and original PMT extracts (two one-sided *t* test). It is generally agreed that if the 90% CI of relative hepatotoxicity compared to original PMT fell within the range of 69.84–143.19% [78], the candidate HEMs were considered to be hepatotoxic equivalent with original PMT extracts. The batches of the S2 PMT sample were chosen to evaluate the hepatotoxic equivalence between candidate HEMs and original PMT extracts. The mixtures of TSG and EG were dissolved in 0.5% CMC-Na solution at a dose equal to that measured in S2 PMT extract. Mice were intraperitoneally injected with 2 mg/kg LPS for 2 h and then sacrificed 6 h after the PMT administration (*n* = 10). The serum ALT and AST activities were used to assess the hepatotoxicity. Meanwhile, the histological samples were also examined.

### 4.9. Hepatotoxic Mechanism of PMT Based on Liver Metabolomics

#### 4.9.1. Preparation of Liver Samples for Metabolomics Analysis

We took 0.30 g of liver tissues into a homogenization tube containing 20 small white magnetic beads, and added 9 times the volume of normal saline to homogenize (S = 5, T = 30 S, Cycle = 3, D = 10 s) to make a turbid solution. We took an appropriate amount of homogenate, centrifuged it (12,000 rpm, 10 min) and collected the supernatant. We precisely aspirated 1 mL of the homogenate supernatant, and added 5 mL of cold acetonitrile, vortexed for 10 min to precipitate proteins, and then continued to centrifuge to get the supernatant. Next, the concentrated supernatants were dissolved with 100 µL 50% acetonitrile solution and then the supernatant was packed in liquid phase vials for analysis. Quality control samples were prepared with equal aliquots by mixing the supernatants of all samples, which were injected every 5 samples to guarantee the stability and repeatability of the UPLC-QTOF/MS systems.

#### 4.9.2. Chromatography and Mass Spectrometry

The Agilent 1290 series UPLC system was used for chromatographic analysis. Samples were separated with an ACQUITY HSS T3 column (2.1 mm × 100 mm, 1.8 μm) at 4 °C. The mobile phase consisted of 0.1% formic acid in water (A) and acetonitrile (B). The conditions of the mobile phase were as follows: 0–4 min, 5–10% B; 4–10 min, 10–50% B; 10–20 min, 50–80% B; 20–25 min, 80–95% B; 25–30 min, 95–95% B; 30–32 min, 95–5% B; 32–35 min, 5–5% B. The column temperature was set at 35 ℃, the injection volume was 10 µL, and the flow rate was 0.3 mL/min.

An Agilent 6550 Q-TOF-MS instrument (Agilent Technologies, Santa Clara, CA, USA) was used. The mass spectrometry analytical conditions were as follows: The electrospray capillary voltage was 3.5 kV in the negative ionization mode and 4.0 kV in the positive ionization mode. The gas temperature was 250 °C and the gas flow was 15 L/min in both negative and positive ionization modes. The nozzle voltage was 1000 V and the mass range was set from *m*/*z* 50 to 1400.

#### 4.9.3. Data Extraction and Analysis

Sample data were extracted using Mass Profiler Professional software (Version13.1, Agilent). The MetaboAnalyst database (Version5.0, https://www.metaboanalyst.ca/, accessed on 13 June 2022) was used to normalize the extracted data, and SIMCA software was used for principal component analysis (PCA) and orthogonal partial least squares discrimination analysis (OPLS-DA). Significant variables (*p* < 0.05 in T test, variable importance for the projection (VIP) > 1.0, and |*p*(corr)| ≥ 0.58) were selected as potential differential metabolites for further pathway enrichment analysis and heatmap analysis in the MetaboAnalyst database.

### 4.10. Construction of an Integrated Network

(1) Collection of liver injury targets: Candidate targets for liver injury were screened through the online database including online Mendelian genetic map (OMIM, https://omim.org/, accessed on 22 June 2022), TTD (therapeutic target database, http://db.idrblab.net/ttd/, accessed on 22 June 2022), TCMSP (Traditional Chinese Medicine Systems Pharmacology Database and Analysis Platform, http://tcmspw.com/tcmsp.php, accessed on 22 June 2022) and Genecards (https://www.gene-cards.org/, accessed on 22 June 2022) by searching keywords of “liver injury” and “hepatotoxicity”.

(2) Computational target fishing of differential metabolites: Five computational tools were used for target fishing of differential metabolites, including PharmMapper (http://www.lilab-ecust.cn/pharmmapper/, accessed on 30 June 2022), SEA (https://sea.bkslab.org/, accessed on 30 June 2022), Stitch (http://stitch.embl.de/, accessed on 30 June 2022), Swiss Target prediction (http://swisstargetprediction.ch/, accessed on 30 June 2022) and Superpred (https://prediction.charite.de/subpages/target_prediction.php, accessed on 30 June 2022).

(3) Collection of metabolic pathway targets: The targets involved in the metabolic pathway were obtained through the KEGG database (https://www.kegg.jp/kegg/pathway.html, accessed on 8 July 2022).

(4) Protein–protein interaction (PPI) and network construction: Through target interactions, the intersections of (1), (2) and (3) were considered as the prediction targets of liver injury caused by PMT.

(5) Protein–protein interactions of common targets were collected from the String database (http://string-db.org/, accessed on 8 July 2022) with a confidence score > 0.4 [99].

(6) The Cytoscape software (version 3.9.1, http://cytoscape.org/, accessed on 8 July 2022) was utilized to construct the networks [100] to build a metabolite–target–metabolic pathway network to visualize the interactions among metabolites, targets and metabolic pathways.

### 4.11. Molecular Docking

Molecular docking was performed based on SYBYL-X 2.0 software to verify the binding force between key metabolites and key targets. We searched and downloaded the structure of each metabolite from the HMDB database (https://hmdb.ca/, accessed on 10 July 2022) and ChemSpider database (http://www.chemspider.com/, accessed on 10 July 2022), then minimized the structures in ChemBio 3D ultra12.0 software. Three-dimensional structures of the target proteins were downloaded from Protein Data Bank database (https://www.rcsb.org/pdb/home/home.do, accessed on 9 July 2022), and SYBYL-X 2.0 software was used to dock metabolites and proteins. The detailed molecular docking operation steps were the same as in a previous work [37]. The total score simulates the binding capacity in the unit of negative logarithm. Therefore, the larger the total score is, the more stable the binding between ligand and receptor is. Generally, from a total score ≥ 5, it can be inferred that there is a strong interaction between the ligand and the corresponding protein target. The larger the total score, the stronger the interactions between the ligand and the corresponding protein target [38].

To analyze the binding affinities and modes of interaction between the drug candidate and their targets, AutodockVina 1.2.2, a silico protein ligand docking software, was employed [101]. The molecular structure of each metabolite was retrieved from the PubChem Compound database (https://pubchem.ncbi.nlm.nih.gov/, accessed on 25 November 2022). The 3D coordinates of target proteins were downloaded from the Protein Data Bank database. For docking analysis, all protein and molecular files were converted into PDBQT format with all water molecules excluded and polar hydrogen atoms added. The grid box was centered to cover the domain of each protein and to accommodate free molecular movement. The grid box was set to 30 Å × 30 Å × 30 Å, and grid point distance was 0.05 nm. The docking program was executed by AutoDock Vina (version 1.1.2, San Diego, CA, USA), and the output score was displayed as kcal/mol.

### 4.12. Detection of Oxidative Stress

We took out the livers to make liver homogenate, and centrifuged to separate the supernatant. Liver homogenate supernatant was used to detect the content of MDA and the activities of SOD and GSH-Px according to the operation steps in the kit instructions. In addition, the protein content was quantified with Bradford kit.

### 4.13. Western Blot

Liver tissues were homogenized in SDS cracking liquid containing protease inhibitors and centrifuged at 15,000 rpm for 10 min to isolate the cell lysate samples. Protein concentration was quantified using Bradford kit. Protein lysate samples were then separated by 10% SDS-PAGE and transferred to a polyvinylidene fluoride membrane at 200 mA for 2 h. The protein expression levels were analyzed using antibodies against NT5E rabbit polyclonal antibody, AK2 rabbit polyclonal antibody, β-actin and anti-rabbit IgG and horseradish peroxidase-linked antibody. Next, the protein band signals were detected with ECL reagent using a chemiluminescence imaging system (ChemiScope Mini, shanghai, China) and quantified by Chemi Analysis software.

### 4.14. Statistical Analysis

The results are expressed as the mean ± standard deviation ($\bar{\chi }$ ± SD). SPSS software (version 17, IBM Corp., Armonk, NY, USA) was applied to analyze data for ANOVA and *p*-values < 0.05 were considered statistically significant.

## 5. Conclusions

Coupled fingerprints with chemometrics, TSG and EG were successfully screened out and verified as the HEMs of PMT. In addition, PMT might cause liver injury by interfering with purine metabolism, inhibiting the protein expressions of NT5E and AK2 and causing oxidative damage and mitochondrial dysfunction. Hopefully, these findings could provide useful guidance about quality control and safety in the study of PMT. Perhaps, considering establishing a reasonable and safe quality control range with HEMs as the quality control index of its medicinal materials and preparations, or reducing the content of HEMs through processing to reduce the toxicity and improve its safety, could be considered in the future.

## Figures and Tables

**Figure 1 molecules-28-00025-f001:**
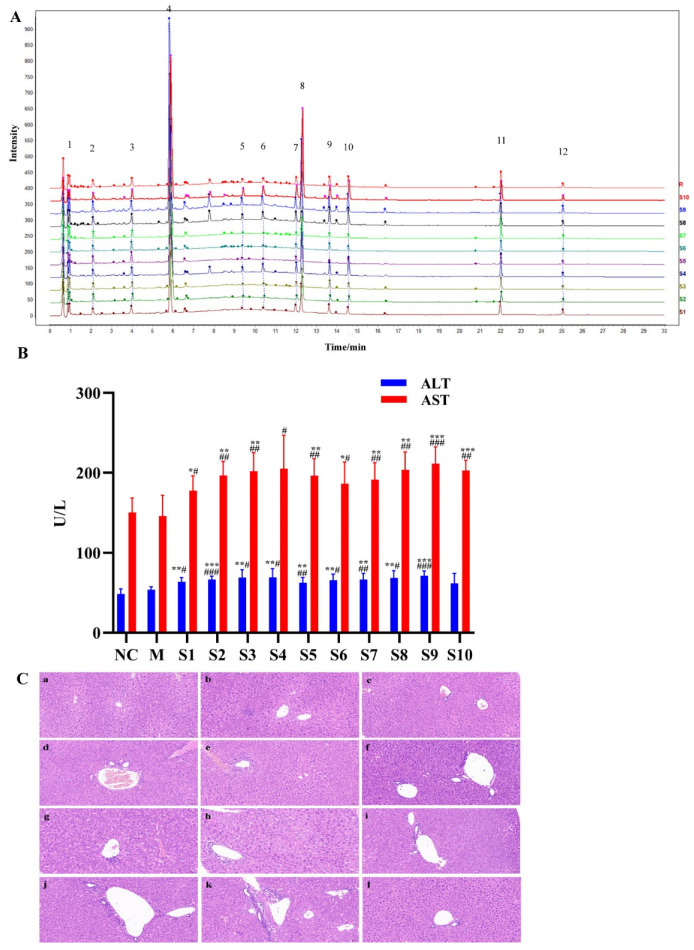
Fingerprints of 10 batches of PMT and the effects on the livers of mice. (**A**) UPLC fingerprints and 12 common peaks of 10 batches of PMT extracts (S01–S10). (**B**) The serum ALT and AST activities of 10 batches of PMT extracts (S01–S10) vs. NC, * *p* < 0.05, ** *p* < 0.01, *** *p* < 0.001; vs. M, ^#^
*p* < 0.05, ^##^
*p* < 0.01, ^###^
*p* < 0.001. (**C**) Histological observation of different batches of PMT extracts-induced liver injury. a, NC; b, M; c. d, e f, g, h, i, j, k and l are representative images showing HE staining of liver tissue from the batch (S1–S10) administration groups, respectively (Ruler: 50 μm).

**Figure 2 molecules-28-00025-f002:**
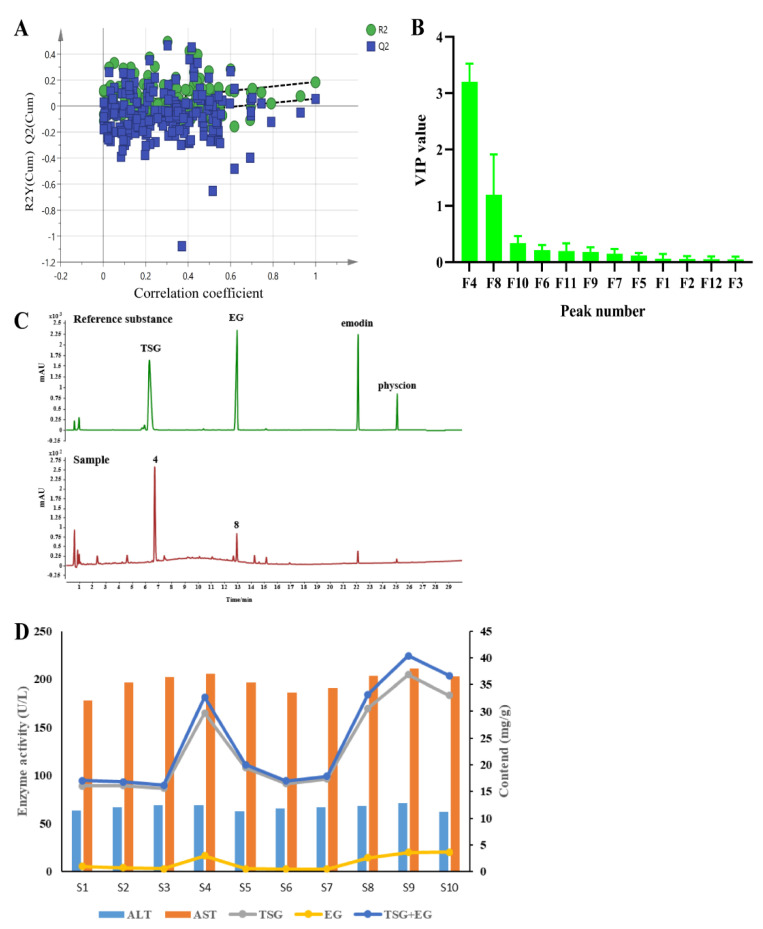
(**A**) Permutation test plot of OPLS model. (**B**) VIP value analysis of variables in OPLS model. (**C**) Chromatogram of reference substance and PMT sample. (**D**) The relationship between the content of TSG and EG and the activities of ALT and AST in 10 batches of PMT (*n* = 10).

**Figure 3 molecules-28-00025-f003:**
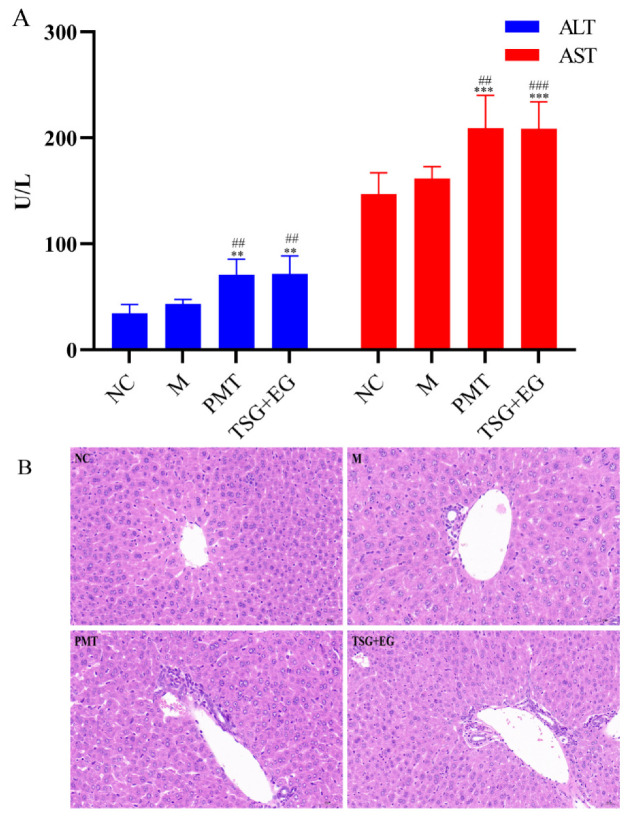
(**A**) Effects of different groups on serum ALT and AST in mice (*n* = 10); vs. NC, ** *p* < 0.01, *** *p* < 0.001; vs. M, ^##^
*p* < 0.01, ^###^
*p* < 0.001. (**B**) Histopathological observations of different groups of induced liver injury (Ruler: 50 μm). NC: normal control group; M: model group; PMT: PMT group.

**Figure 4 molecules-28-00025-f004:**
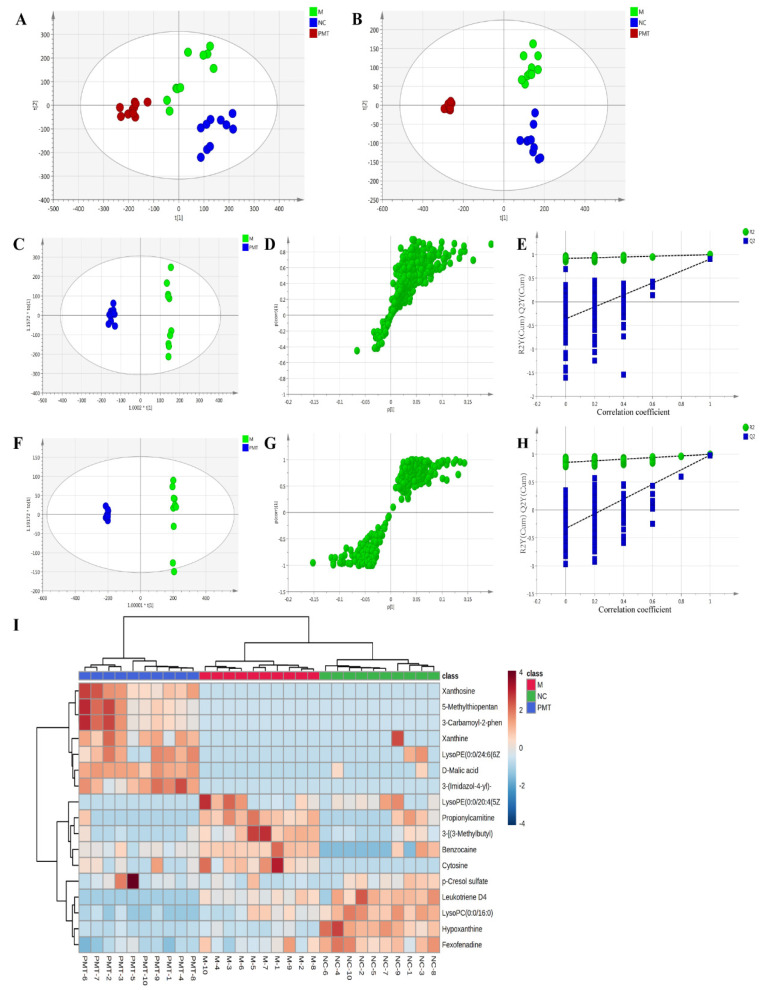
(**A**,**B**) PCA diagram under positive and negative ion modes. (**C**,**F**) OPLS analysis diagram of liver samples in each group under positive and negative ion modes. (**D**,**G**) S−plot diagram under positive and negative ion modes. (**E**,**H**) The validation plots. (**I**) Heat diagram between 17 potential biomarkers. NC: normal control group; M: model group; PMT: PMT group.

**Figure 5 molecules-28-00025-f005:**
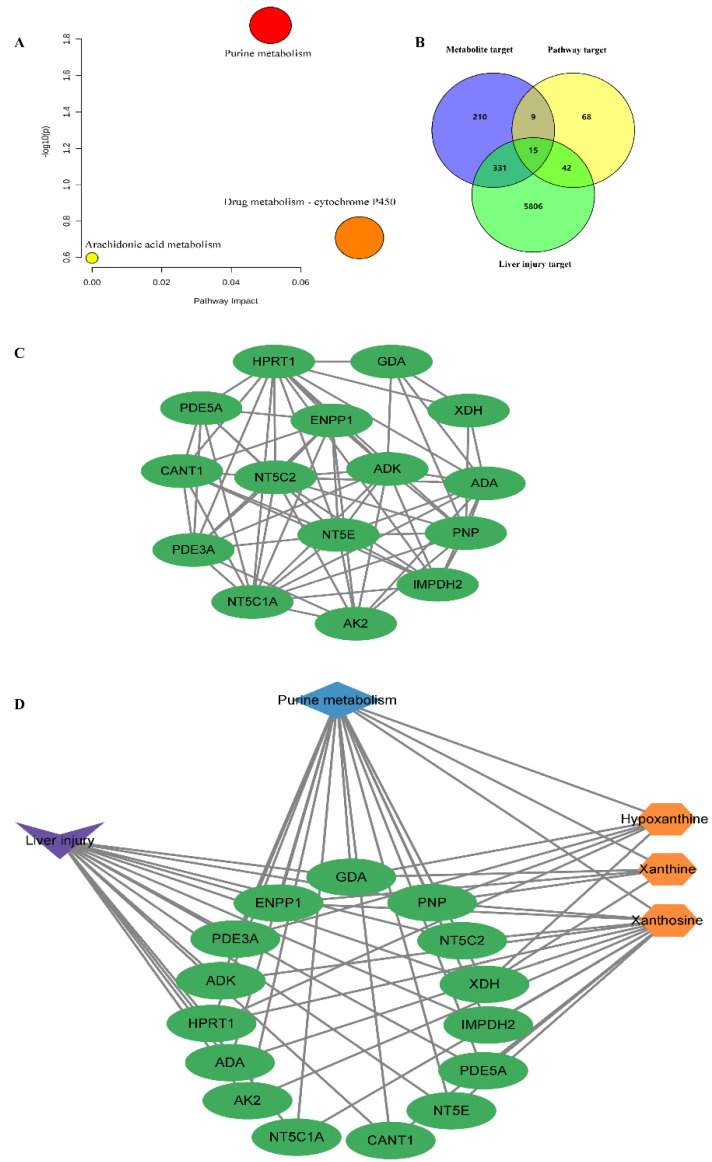
(**A**) KEGG pathway enrichment map of metabolites; (**B**) Venn diagram; (**C**) interactions between the 15 overlapping key targets; (**D**) metabolites, overlapping key targets and metabolic pathways related to purine metabolism in PMT.

**Figure 6 molecules-28-00025-f006:**
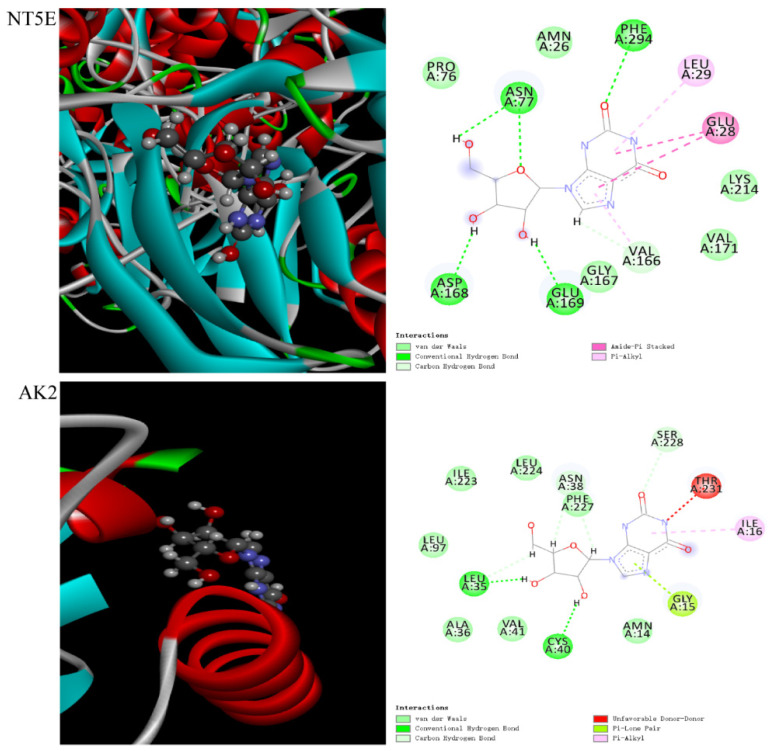
Schematic diagram of molecular docking between xanthosine and target NT5E, as well as AK2.

**Figure 7 molecules-28-00025-f007:**
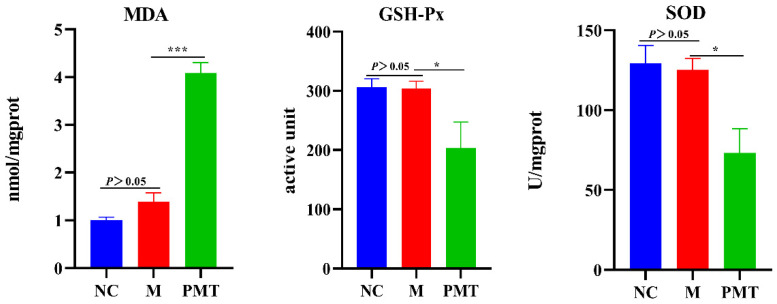
Effects of PMT on MDA content, SOD and GSH-Px activities; vs. M, * *p* < 0.05, *** *p* < 0.001. NC: normal control group; M: model group; PMT: PMT group.

**Figure 8 molecules-28-00025-f008:**
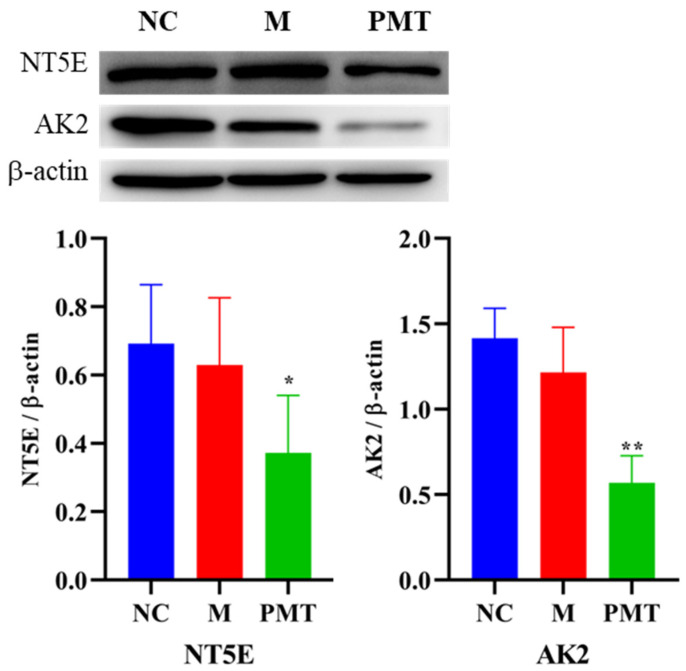
Effects of PMT on protein expressions of NT5E and AK2; vs. M, * *p* < 0.05, ** *p* < 0.01. NC: normal control group; M: model group; PMT: PMT group.

**Figure 9 molecules-28-00025-f009:**
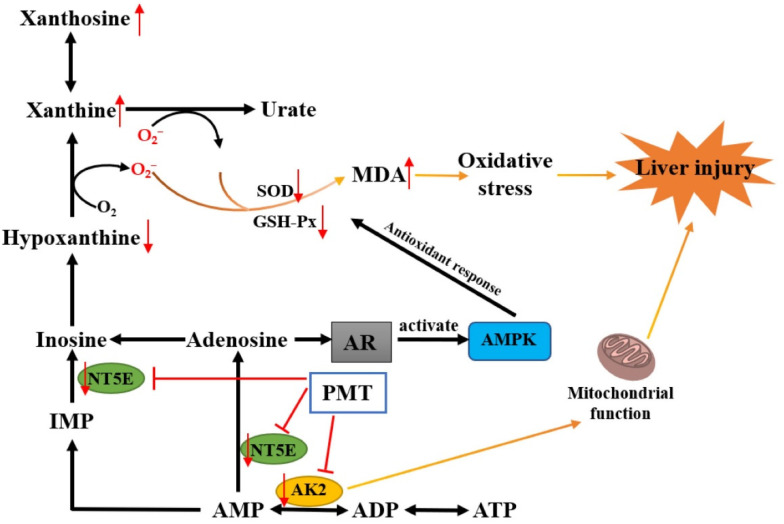
A schematic diagram of liver injury caused by PMT via purine metabolism.

**Table 1 molecules-28-00025-t001:** Fingerprint similarity analysis of 10 batches of PMT samples.

	S1	S2	S3	S4	S5	S6	S7	S8	S9	S10	Reference Fingerprint
S1	1.000	0.995	0.993	0.983	0.99	0.992	0.985	0.986	0.975	0.967	0.992
S2	0.995	1.000	0.998	0.981	0.995	0.992	0.992	0.984	0.977	0.958	0.992
S3	0.993	0.998	1.000	0.98	0.993	0.992	0.995	0.982	0.975	0.95	0.99
S4	0.983	0.981	0.98	1.000	0.979	0.976	0.971	0.997	0.997	0.987	0.998
S5	0.99	0.995	0.993	0.979	1.000	0.992	0.991	0.982	0.976	0.95	0.989
S6	0.992	0.992	0.992	0.976	0.992	1.000	0.988	0.979	0.971	0.951	0.987
S7	0.985	0.992	0.995	0.971	0.991	0.988	1.000	0.974	0.965	0.937	0.982
S8	0.986	0.984	0.982	0.997	0.982	0.979	0.974	1.000	0.995	0.984	0.998
S9	0.975	0.977	0.975	0.997	0.976	0.971	0.965	0.995	1.000	0.979	0.994
S10	0.967	0.958	0.95	0.987	0.95	0.951	0.937	0.984	0.979	1.000	0.982
Reference fingerprint	0.992	0.992	0.99	0.998	0.989	0.987	0.982	0.998	0.994	0.982	1.000

**Table 2 molecules-28-00025-t002:** Spearman correlation analysis between the contents of TSG, EM, TSG + EG and serum ALT and AST activities in mice in 10 batches of PMT.

	Group	Spearman r	*p* Value
	TSG	0.3043	0.3926
ALT	EG	0.2467	0.492
	TSG + EG	0.2973	0.4041
	TSG	0.7552	0.0115
AST	EG	0.6868	0.0283
	TSG + EG	0.7482	0.0128

**Table 3 molecules-28-00025-t003:** Differential metabolites in the livers of mice affected by PMT.

No	Retention Time (min)	*m*/*z*	Ionization Mode	Formula	Metabolites	HMDB ID	PMT/ M	KEGG Pathway
1	1.1	134.0230	M − H	C_4_H_6_O_5_	D-Malic acid	HMDB0031518	up	Butanoate metabolism
2	1.2	220.0492	M − H	C_6_H_9_N_2_O_5_P	3-(Imidazol-4-yl)-2-oxopropyl phosphate	HMDB0012236	up	Histidine metabolism
3	1.2	284.0787	M − H	C_10_H_12_N_4_O_6_	Xanthosine	HMDB0000299	up	Purine metabolism
4	1.6	136.0372	M + H	C_5_H_4_N_4_O	Hypoxanthine	HMDB0000157	down	Purine metabolism
5	1.9	152.0348	M − H	C_5_H_4_N_4_O_2_	Xanthine	HMDB0000292	up	Purine metabolism
6	2.6	217.1327	M + H	C_10_H_19_NO_4_	Propionylcarnitine	HMDB0000824	down	/
7	3.7	165.0793	M + H	C_9_H_11_NO_2_	Benzocaine	HMDB0004992	down	/
8	6.6	495.3236	M + H	C_24_H_50_NO_7_P	LysoPC(0:0/16:0)	HMDB0240262	down	/
9	6.8	186.1358	M + H	C_9_H_18_N_2_O_2_	3-[(3-Methylbutyl)nitrosoamino]-2-butanone	HMDB0033553	down	/
10	6.8	501.2864	M + H	C_32_H_39_NO_4_	Fexofenadine	HMDB0005030	down	/
11	6.8	111.0432	M + H	C_4_H_5_N_3_O	Cytosine	HMDB0000630	down	Pyrimidine metabolism
12	8.0	147.0764	M − H	C_6_H_13_NOS	5-Methylthiopentanaldoxime	METPA1772	down	Glucosinolate biosynthesis
13	8.4	188.0184	M − H	C_7_H_8_O_4_S	p-Cresol sulfate	HMDB0011635	up	Toluene degradation
14	8.5	193.0755	M − H	C_10_H_11_NO_3_	3-Carbamoyl-2-phenylpropionaldehyde	HMDB0060366	up	Drug metabolism—cytochrome P450
15	9.2	501.2844	M − H	C_25_H_44_NO_7_P	LysoPE(0:0/20:4(5Z,8Z,11Z,14Z))	HMDB0011487	down	/
16	15.2	501.294	M − H	C_25_H_35_D_5_N_2_O_6_S	Leukotriene D4	HMDB0003080	down	Arachidonic acid metabolism
17	15.6	553.3491	M − H	C_29_H_48_NO_7_P	LysoPE(0:0/24:6(6Z,9Z,12Z,15Z,18Z,21Z))	HMDB0011499	up	/

**Table 4 molecules-28-00025-t004:** Relevant information of three metabolic pathways obtained by KEGG enrichment.

Pathway Name	Match Status	*p*	−log(p)	FDR	Impact
Purine metabolism	3/66	0.013319	1.8755	1	0.05133
Drug metabolism–cytochrome P450	1/27	0.19579	0.70821	1	0.07692
Arachidonic acid metabolism	1/36	0.25279	0.59723	1	0

**Table 5 molecules-28-00025-t005:** Molecular docking results determined by SYBYL-X.

Metabolite	Protein	Total Score	Crash	Polar	D Score	PMF Score	G Score	Chem Score	Global CScore	Similarity
Hypoxanthine	NT5E	3.62	−0.46	1.99	−35.96	−24.68	−109.77	−13.56	2	0.44
	XDH	3.68	−0.40	4.81	−21.21	−29.74	−54.39	−15.96	2	0.11
	PDE3A	4.58	−0.06	3.91	−41.96	−18.26	−79.18	−6.62	1	0.44
	PDE5A	4.79	−0.47	3.54	−61.71	−28.38	−119.14	−7.23	2	0.45
	ADK	3.43	−0.32	0.44	−61.35	−21.57	−127.56	−7.29	2	0.46
	ENPP1	3.21	−0.28	3.32	−50.70	−27.29	−83.53	−6.55	1	0.32
	ADA	3.45	−0.22	1.79	−49.32	−28.60	−94.28	−9.73	L	0.19
	PNP	3.59	−0.21	0.92	−60.20	−18.68	−106.95	−8.91	2	0.43
	NT5C2	3.42	−0.79	2.00	−71.08	−55.92	−113.60	−14.40	2	0.40
	AK2	4.99	−0.14	4.08	−49.17	−19.32	−84.25	−6.68	2	0.46
	IMPDH2	2.70	−1.29	1.77	−54.66	−21.21	−118.69	−13.58	2	0.40
	CANT1	4.17	−0.37	3.51	−54.69	−33.65	−83.43	−12.94	1	0.29
	HPRT1	3.68	−0.05	2.68	−31.02	−50.60	−74.85	−13.58	1	0.21
	GDA	3.12	−0.09	4.08	−11.87	−14.64	−42.34	−6.16	2	0.37
Xanthosine	NT5E	6.54	−0.60	5.39	−107.22	−44.53	−204.37	−5.17	4	0.51
	XDH	1.77	−1.01	2.57	−44.63	−5.37	−77.36	−6.51	2	0.18
	PDE3A	4.77	−1.48	3.59	−125.93	−1.01	−140.46	−7.95	2	0.29
	PDE5A	−4.43	−11.72	2.45	−156.64	26.82	−255.74	−13.94	2	0.52
	ADK	−4.09	−12.85	1.40	−183.09	39.70	−258.38	−5.82	2	0.39
	ENPP1	4.89	−0.53	3.71	−97.29	−52.72	−126.75	−6.24	2	0.42
	ADA	3.31	−4.25	1.67	−135.84	−34.69	−225.05	−9.18	2	0.42
	PNP	−1.60	−8.85	2.78	−135.42	28.43	−233.08	−11.94	2	0.36
	NT5C2	4.27	−5.60	6.40	−170.44	−129.21	−200.58	−15.48	2	0.51
	AK2	5.62	−0.98	4.25	−94.40	−9.20	−141.24	−5.76	3	0.50
	IMPDH2	0.21	−5.93	2.14	−155.75	15.64	−217.59	−12.97	2	0.51
	CANT1	4.00	−1.91	2.30	−140.67	−57.79	−165.05	−3.05	2	0.32
	HPRT1	4.40	−0.98	4.69	−100.57	−36.89	−146.16	−14.66	1	0.43
	GDA	3.43	−0.84	2.98	−88.25	−17.53	−155.70	−4.31	3	0.49
Xanthine	NT5E	3.42	−0.31	2.17	−48.57	−17.36	−111.29	−12.89	2	0.40
	XDH	1.82	−0.16	2.68	−32.42	−35.69	−46.75	−12.65	2	0.14
	PDE3A	4.12	−0.18	3.56	−55.08	−16.02	−95.64	−9.92	1	0.44
	PDE5A	4.42	−1.06	3.14	−79.06	−12.36	−129.64	−7.10	2	0.42
	ADK	3.63	−0.10	0.02	−76.60	−25.27	−131.98	−9.44	2	0.48
	ENPP1	4.70	−0.09	4.57	−50.32	−29.71	−65.72	−12.92	3	0.28
	ADA	3.36	−0.53	1.41	−77.80	−62.12	−127.63	−13.13	2	0.47
	PNP	4.34	−0.22	1.98	−71.61	−3.81	−118.35	−6.92	2	0.58
	NT5C2	3.08	−1.69	2.80	−76.81	−63.86	−117.95	−12.74	2	0.48
	AK2	3.08	−0.11	2.13	−50.38	−16.15	−83.59	−6.73	1	0.37
	IMPDH2	2.50	−2.45	2.21	−74.40	1.40	−134.67	−11.39	2	0.47
	CANT1	4.37	−0.48	4.41	−62.35	−28.46	−94.41	−12.13	1	0.29
	HPRT1	3.47	−0.71	3.28	−52.21	−32.82	−81.77	−9.90	1	0.46
	GDA	2.12	−0.06	2.49	−32.17	−16.37	−55.93	−9.41	1	0.17

**Table 6 molecules-28-00025-t006:** Molecular docking results given by AutoDock.

Metabolite	Protein	Binding Energy (kcal/mol)
Xanthosine	NT5E	−6.95
	AK2	−6.58

## Data Availability

Data used to support the findings of this study are available from the corresponding author upon request.

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
