# Peer review of "Discovery of Hepatotoxic Equivalent Markers and Mechanism of Polygonum multiflorum Thunb. by Metabolomics Coupled with Molecular Docking"

_molecules, 2022, doi:10.3390/molecules28010025_

Round 1

Reviewer 1 Report

In the manuscript Discovery of Hepatotoxic Equivalent Markers and Mechanism of Polygonum Multiflorum Thunb. by Metabolomics coupled with Molecular Docking, the authors have attempted to elaborate the hepatotoxic equivalent markers of Polygonum multiflorum Thunb. (PMT), a commonly used Chinese herbal medicine but with potential liver injury effects. They have provided good statistical and docking data to establish the claim. However, the following points must be taken into consideration to increase the likelihood of acceptance of the paper.

·         PMT affects purine metabolism by targeting NT5E, AK2 to induce liver injury. Performing a qPCR and western blot would help solidify the claim and is suggested in this regard. Also, to confirm the claim that PMT might affect those by upregulating xanthosine.

·         Providing a separate paragraph on which genes are downregulated/up-regulated in liver toxicity by PMT is suggested.

·         The authors are suggested to give a correlation between liver toxicity level and with dose of PMT. And can this toxic nature be repurposed for cytotoxicity studies?

·         The authors claimed that finding out the HEMs is beneficial in a way that it can be put into safe clinical application. But how? The take-home message is not clear and should be properly elaborated in the conclusion part.

·         The figures, network diagrams, as well as pathways are not crisp and clear. It is suggested to provide good quality images with the proper size. Also, as per the author’s claims based on their results, PMT engages in the purine metabolism pathway and obstructs it. It is highly suggested to provide a schematic pathway diagram and show where PMT is acting upon.

·         It is claimed in the discussion that the upregulation of purine-related metabolites likely increases ROS. So, analyzing ROS by performing ROS assays would provide a clear view of how PMT is hampering the hepatocytes and playing a role to obstruct the purine metabolism pathway.

·         In line no. 129- D doesn’t have a bracket. Line no. 138, 141 and 144 what is EM? In line no. 146 what is HECM ? and in line no. 13 Polygonum multiflorum Thunb would be in italics. Also, in the title of the manuscript, the spelling of “thunb” is written as “thumb”. The suffix thunb is given after naturalist Carl Peter Thunberg who was a student of Carl Linnaeus.

Author Response

I am very grateful for your comments on the manuscript. I have provided an attachment with my reply. Please download and check it. Thank you again!

Reviewer 2 Report

Title;Discovery of Hepatotoxic Equivalent Markers and Mechanism of Polygonum Multiflorum Thumb. by Metabolomics coupled with Molecular Docking
Comments;In my view, the results obtained in this study are worthy for publication. The manuscript needs major essential revision before publication. I would like to overview the revised version of the manuscript. I have the following comments/suggestions for authors to address before final decision on the manuscript.
1. Rationale behind the selection of threshold value “Metabolites that meet a threshold of p < 0.05, VIP > 1.0 and |p(corr)| ≥ 0.58”
2. In figure 2 authors have suggested to rewrite the axis titles to increase the visibility of the graph.
3. Provide the reference for the statement “MetaboAnalyst 5.0 was applied to reveal the metabolic pathways of potential metab- 198olites related to PMT.”
4. Authors have advised to provide the gist of the crash written in table 5 and their advantage in the context of the docking score of molecular docking analysis.
5. Rewrite the sentence “The hepatotoxic equivalence between candidate HEMs and PMT extracts was exactly validated in mice.”
6. In the Introduction section the author should refer to the research paper and comment on recent in-silico techniques. It will be good information for the readers. I would like to recommend several papers, among many others, providing further explanation on this topic: PMID: 35362492 PMID: 27194485 PMID: 32619131 PMID: 35276295 PMID: 34299222 PMID: 23383724
7. “Polygonum multiflorum Thunb. (PMT), a commonly used Chinese herbal”: Punctuation error.
8. Avoid usage of non-standard abbreviations such as “e.g.”, etc. in the manuscript.
9. In vivo should be italicized throughout the manuscript.
10. Define “AK2” and “NT5E” in the Abstract.
11. Clearly define the aim and objectives of the study in the last paragraph of the Introduction section.
12. “The details were the same as previous work[37].” What details are the authors talking about?
13. How the binding sites were defined on each protein?
14. Provide citations to the PDB IDs mentioned in the manuscript. Also, were the criteria for the selection of structures for docking studies?
15. “4.2. Preparation of PMT extracts” Authors should provide information about the overall extract yield in the result section.
16. Authors have performed the molecular docking of 3 compounds with so many targets. They did not use any standard on how comparisons can be made.
17. Authors need to validate the docking protocol.
18. “Figure 6. Schematic Diagram of Molecular Docking between xanthosine and target NT5E, AK2.” There is one unfavorable interaction in the figure. How authors justify this.

Author Response

(The authors gave the same response as above.)

Round 2

Reviewer 1 Report

Figure 5 (C) still looks a bit hazy and unclear. It is recommended to replace it with a higher-resolution version. 

Author Response

Thank you for your suggestion one more time. We have adjusted the resolution of Figure 5 to 600 dpi (line252). Figure 5 (C) is a network diagram of the relationships among 15 key targets. The Figure 5 (C) provided previously was downloaded directly from the string platform and might look unclear. Therefore, we have imported the relevant data of 15 targets into cytoscape software, redrawn the images, and output the image Figure 5 (C) with a resolution of 600 dpi. Subsequently, we have integrated Figure 5A, B, C, D through Ai software and exported it to Figure 5 of 600 dpi.

Reviewer 2 Report

The authors have responded to all concerns meticulously and improved the manuscript accordingly. The revised draft is improved significantly. I do not have further comments.

Author Response

Thank you very much for your comments. Thank you for your thoughtful guidance before, so that the manuscript has made significant improvement. Once again, we would like to express our heartfelt thanks for your suggestions.